# A Crowded Object Counting System with Self-Attention Mechanism

**DOI:** 10.3390/s24206612

**Published:** 2024-10-14

**Authors:** Cheng-Chang Lien, Pei-Chen Wu

**Affiliations:** Department of Computer Science & Information Engineering, Chung Hua University, Hsinchu City 300110, Taiwan; m10902016@ms.chu.edu.tw

**Keywords:** crowded object counting, density map, self-attention mechanism

## Abstract

Traditional object counting systems use object detection methods to count objects. However, when objects are small, crowded, and dense, object detection may fail, leading to inaccuracies in counting. To address this issue, we propose a crowded object counting system based on density map estimation. While most density map estimation models employ encoder–decoder or multi-branch approaches to generate feature maps at different scales for obtaining an accurate density map, improving the accuracy of crowded object counting remains a challenge. In this paper, we propose a novel model that can generate more accurate density maps, utilizing the context-aware network as the primary structure and integrating the self-attention mechanism. There are three main contributions in this paper. Firstly, the self-attention mechanism is employed to improve the accuracy of density map estimation. Secondly, the missing vehicle labels in the TRANCOS database are relabeled, ensuring that the ground truth data are more complete than the original TRANCOS database, thus enabling the proposed novel model to have higher crowded object counting accuracy. Thirdly, the parameters of the self-attention mechanism are analyzed to obtain the optimum parameter combination. The experimental results demonstrate that the accuracy of crowded object counting can reach 85.9%, 90.0%, 83.4%, and 92.6% for the TRANCOS, relabeled TRANCOS, ShanghaiTech Part A, and Part B datasets, respectively. Furthermore, the ablation study for the context-aware network with self-attention mechanism analyzes the optimum parameter combination.

## 1. Introduction

The object counting system has various potential applications, including evacuation systems, vehicle detection and traffic management, and security at athletic events. Sabeur et al. [1] proposed a vision-based evacuation system that analyzes dense crowd behavior in confined environments such as subway stations, football stadiums, cruise liners, and international airports. Loakeim et al. [2] introduced an anticipatory system designed to operate during pedestrian evacuations, preventing congestion at escape points. Zehang et al. [3] reviewed several robust and reliable vision-based vehicle detection and traffic management systems for roadways. In [4], a people-counting technology for sports venues was proposed to address severe occlusion issues by using a deep model-based approach that acts as a head detector and accounts for variations in head scale within videos.

Most traditional object counting systems use object detection methods for counting. Teoh et al. [5] locate vehicles by utilizing an edge oriented histogram (EOH) to extract the symmetrical vehicle edge features and then employ the support vector machine (SVM) to detect vehicle positions. Pham et al. [6] locate vehicles based on the features of the vehicle’s windshield frame. Although these methods can detect the objects under slight occlusions, they fail in situations where objects are small and densely crowded.

Girshick et al. proposed R-CNN [7], the first model to introduce CNN into the field of object detection. R-CNN first uses the selective search method to find candidate object regions and then uses a CNN to obtain object features. The same authors later improved R-CNN and proposed Fast R-CNN [8], which not only improved the network architecture to an end-to-end structure but also increased object detection accuracy. However, Fast R-CNN still required a significant amount of time and parameters to extract 2000 candidate regions. Later, Girshick et al. proposed Faster R-CNN [9], which uses a region proposal network (RPN) to eliminate the need to extract 2000 candidate regions, thus saving a lot of computation time and improving object detection accuracy. Fan et al. [10] achieved a 95.14% vehicle detection accuracy on the KITTI vehicle dataset using the faster R-CNN network. Another well-known object detection series is the YOLO object detection series, where studies [11,12,13,14,15] are based on the YOLOv4 [16] model to optimize vehicle detection accuracy. Furthermore, an artificial-intelligence-enabled IIoT-based framework with VD-Network (VD-Net) [17] is proposed to detect suspicious objects intelligently.

Nevertheless, the YOLO-based vehicle detection methods are only suitable for situations where the vehicle density is relatively sparse. In cases where traffic congestion results in vehicles being densely packed and heavily occluding each other, as illustrated in Figure 1a, or when vehicles are too small due to their distance from the road camera, as shown in Figure 1b, traditional object detection methods struggle to detect the vehicle regions. Additionally, significant variations in lighting conditions, viewing angles, and changes in vehicle distribution and density within the scene can decrease the effectiveness of object detection.

Facing this challenge, the traditional object detection methods prove less than ideal for achieving precise object counting in crowded object scenes. Instead of accurately detecting the position, contours, and size of each object to draw bounding boxes, the crowded object counting becomes more suitable to generate a Gaussian density map using only position and quantity information. This shift in approach allows for the creation of a more fitting training target for the task of precise object counting in the image.

## 2. Related Works

The generation of density maps involves first manually marking the center points of objects. Then, based on the coordinates of these center points, the area of the object is transformed into a probability distribution using a Gaussian kernel. Therefore, the closer to the object center, the higher the probability value, and vice versa, the farther away from the object, the lower the probability value. The sum of the area values for an object is equal to 1. The value of the standard deviation σ affects the probability density distribution of the object. If the standard deviation is smaller, the probability density distribution is more concentrated, and the coverage range is smaller. Conversely, if the standard deviation is larger, the probability density distribution is more dispersed. By using this method with each object’s coordinates as the center point, performing Gaussian kernel operations one by one, we can generate probability density distribution maps of densely populated objects. Summing the entire density map yields the number of objects in that density map. The advantage of density maps is that they not only provide information about the total number of objects but also help avoid the problem of object detection failure in densely populated areas.

Tayara et al. [18] designed a fully convolutional regression network (FCRN) based on density map regression to locate and count vehicles in aerial photographs. They used a design similar to an autoencoder to obtain features of different scales to address the problem of significant variations in vehicle sizes in images. Oñoro-Rubio et al. [19] proposed the Hydra CNN model to ensure uniform density across all blocks. They achieve this by using blocks of different sizes to capture regions of different sizes in the image, then scaling them to a uniform size. They then use a multi-branch approach to learn features at different scales separately. Finally, they merge features of different scales using a set of fully connected layer to output the density map.

In Ref. [20], Sam et al. proposed the switching CNN model that divides the image into nine equally sized patches and uses a pre-trained classifier to differentiate the density levels of different patches. Based on their density levels, the patches are then input into different branches for training. This approach allows the model to learn features at multiple scales. However, the additional pre-training step required to distinguish different density levels makes the process more complex and multi-staged.

Lin et al. [21] introduced a multifaceted attention network (MAN) to enhance transformer models for crowd image analysis by encoding local spatial relations to address the problem of large-scale variations that often exist within crowd images. Han et al. [22] proposed a novel method called STEERER to address scale variations in object counting. STEERER optimizes feature extraction by selecting the best scale for patch objects and progressively inheriting only discriminative features from lower to higher resolutions. Ranasinghe et al. [23] proposed a CrowdDiff crowd counting system in which the crowd density maps are generated through a reverse diffusion process to enhance feature learning, thereby improving crowd counting performance. Wang et al. [24] utilized a diffusion model to generate extensive smoothed density maps used as the training data to significantly improve ControlNet’s [24] performance in crowd counting. However, the training cost of the diffusion model is high.

To overcome the large-scale variations within crowd images, Liu et al. [25] proposed the context-aware network architecture, which is based on multi-scale branches and can be trained end-to-end. This model utilizes global average pooling layers to obtain feature maps with multiple receptive fields of different scales and merges them, enabling adaptive learning of features at different scales. This is also the primary reference for this paper. Yang et al. [26] incorporated spatial and channel-wise attention mechanisms into the vehicle density map estimation network, effectively improving the accuracy of vehicle counting without significantly increasing parameters. This discovery highlights the effectiveness of attention mechanisms. Therefore, this paper chooses to use the self-attention mechanism [27] as a feature enhancement method. The application of the self-attention mechanism has become popular in computer vision since the proposal of the vision transformer method [28]. Self-attention allows each pixel in an image to attend to all other pixels, making it effective for image classification tasks and comparable to state-of-the-art CNN models.

This paper combines the context-aware network architecture with the self-attention mechanism of the vision transformer to propose a new model for counting crowded and dense objects, as illustrated in Figure 2. It aims to make three research contributions:Improvement of Density Map Prediction: The utilization of the self-attention mechanism from the vision transformer is integrated to the context-aware network for improving the accuracy of the density map prediction of vehicle clusters and crowds.Reannotation of TRANCOS Database: Manual reannotation of vehicle coordinates that were previously overlooked in the TRANCOS database is performed. This ensures accurate predictions of the number of vehicles in the TRANCOS database.Study of Optimum Self-Attention Parameters: A study is conducted on various parameters in combining self-attention modules to analyze their impact on accuracy and identify the optimal parameter combination. This analysis is crucial for enhancing the performance of the context-aware network when integrating self-attention modules.

The proposed contributions are anticipated to advance the state-of-the-art in object counting models, particularly in scenarios with dense and crowded objects.

## 3. Context-Aware Network with Self-Attention Mechanism

The goal of this study is to accurately predict the number of objects in scenes with extreme congestion, overcoming challenges such as changes in perspective, object distribution, and density. To address these challenges, the solution lies in enabling the model to learn multi-scale features to adapt to different object sizes. The proposed approach in this paper integrates the self-attention mechanism into the context-aware network for improving counting accuracy.

### 3.1. Context-Aware Network

In the context-aware network, the front-end network first extracts image features from the input image. According to [29], it is observed that the use of the first ten layers of the VGG-16 yields the best feature extraction results. The key module in the context-aware network is the multiscale feature difference fusion module [25] shown in Figure 3. The input image undergoes a total of 10 convolutional layers and 3 max-pooling layers to obtain the tensor ***F_v_***. After obtaining the features extracted by the front-end network from the input image, a multiscale feature difference fusion module is employed to perform multiscale feature extraction and fusion. This is performed to capture features of objects with varying sizes.

VGG16, as the front-end network for feature extraction, has a drawback in that it learns on the same receptive field. To address this limitation and enable the learning of multiscale feature differences, the input feature map undergoes global average pooling to transform it into four different resolution branches: 1 × 1, 2 × 2, 3 × 3, and 6 × 6. Following that, upsampling is applied to restore the feature map to its original size. This process not only brings the feature map back to its original dimensions but also has a smoothing effect on the boundaries between blocks, making the separation lines less obvious. The feature map at this stage is depicted in Figure 4, and the process is defined by Equation (1).
(1)$$S_{j}=U_{bi}(F_{j}({GP}_{ave}\left(F_{v},j\right),\theta_{j}))$$

The notation *j* ranging from 1 to 4 represents the different scales. *GP_ave_*(⋅,*j*) denotes the global average pooling layer, which transforms the input feature map into a size of *k*(j) × *k*(j), where *k*(j) ∈ {1, 2, 3, 6}. ***F_j_***(⋅, *θ*_j_) represents a 1×1 convolutional layer with parameters ***θ****_j_* corresponding to different scales. U_bi_ denotes the bilinear interpolation for upsampling, aiming to restore the feature map to the same size as ***F_v_***. Then, the upsampled feature map ***S****_j_* is obtained. The subtracted difference feature map ***C****_j_* is derived by subtracting ***S****_j_* from the original ***F_v_***, revealing differences in various regions. This process facilitates learning differences between specific features and those in their surrounding areas. This is defined by Equation (2).
(2)$$C_{j}=S_{j}-F_{v}$$

Then, the 1 *×* 1 convolution layer is used again for feature enhancement. At this point, the Sigmoid function is applied to perform feature value transformation. This process can generate the weighting feature map for each specific scale under different branches, as defined in Equation (3).
(3)$$W_{j}={Sig(Conv}^{j}(C_{j},\theta^{j}))$$

*Conv*() represents the 1×1 convolution layer and *Sig*() denotes the Sigmoid function for different scales in different branches, $\theta^{j}$ represents the parameters required for the convolution layer, and ***W****_j_* represents the weighted feature maps for different scales. Then, on each branch, the weighted map ***W****_j_* is element-wise multiplied with ***S****_j_* obtained earlier to yield the weighted regional difference feature map. Finally, the feature maps from all branches are summed to obtain the final output feature map, as shown in Equation (4).
(4)$$F_{d}=\sum_{j=1}^{x}W_{j}⊙S_{j}$$

Here, ⊙ denotes element-wise multiplication between ***W***_j_ and ***S***_j_. *x* represents the number of branches, which is 4 in this case. ***F****_d_* represents the output weighted difference feature map. After obtaining the weighted difference feature map, in order to enhance spatial information and improve the accuracy of object position prediction, a residual connection is used to concatenate the ***F****_v_* just processed by VGG16 with ***F****_d_*, as defined in Equation (5).
(5)$$f_{concat}=\left[F_{v}|F_{d}\right]$$

Here, [∙|∙] is used to denote concatenation of two feature maps along the channel dimension. Since both ***F***_v_ and ***F***_d_ have a depth of 512, the depth becomes 1024 after concatenation. The concatenated feature map is then input into the density map output block, where a series of convolutional layers are used for feature extraction and gradual dimension reduction to obtain the final crowded object density map. The loss function used for training this model is the function of mean square error (MSE), as defined in Equation (6). The optimizer used for model parameter optimization is Adam, as described in [30].
(6)$$MSE(\theta )=\frac{1}{2B}\sum_{i=1}^{B}{\left\|D_{i}^{gt}-D_{i}^{est}\right\|}_{2}^{2}$$
where *B* represents batch size, ***D****_i_*^gt^ represents the ground truth of vehicle density map, and ***D****_i_*^est^ represents the object density map predicted by the model.

### 3.2. Integration of Context-Aware Network and Self-Attention Mechanism

This paper combines the context-aware network architecture with the self-attention mechanism of the vision transformer to propose a new crowded object counting model (as shown in Figure 5). Here, the multi-head attention mechanism [31] shown in Figure 6 is applied to enhance the feature tensor ***F***_d_ for generating the feature tensor ***F****_SA_*. The multi-head attention module is a critical component in transformer architectures, like those used in models such as BERT [32] and GPT [33]. Its primary function is to enhance the model’s ability to focus on different parts of the input sequence in parallel, thereby capturing a wide range of relationships and dependencies. Instead of computing a single attention function, multiple attention heads (each with different sets of parameters) are computed in parallel. Each head produces a different representation, allowing the model to capture various aspects of the input sequence.

The self-attention mechanism converts the feature map into multiple one-dimensional vectors. To accelerate training speed, mitigate gradient vanishing and explosion issues, etc., the feature map undergoes layer normalization [34], normalizing the feature vectors ***v***_n_. Subsequently, the resulting vectors can be inputted into the multi-head self-attention module, as depicted in Figure 6. The multi-head attention method can acquire more than one type of correlation between vectors, and there could be many different types of correlations simultaneously present. Therefore, the multi-head attention module subdivides the previously mentioned ***q***_n_, ***k***_n_, ***v***_n_ by multiplying them with matrices ***W***_q,i_, ***W***_k,i_, and ***W***_v,i_, into multiple ***q***_n,i_, ***k***_n,i_, ***v***_n,i_, where *n* depends on the total number of vectors, and *i* depends on the number of heads, as illustrated in Figure 6.

After obtaining the attention weight values through the multi-head attention mechanism, they are added to the initial input through residual connections, as defined by Equation (7).
(7)$${SA}_{n}=MHA\left(LN\left(v_{n-1}\right)\right)+v_{n},n=1\ldots N$$
where *MHA*() is the multi-head self-attention function. Let *SA*_n_ denote the vectors obtained after the multi-head attention block. Subsequently, all these one-dimensional vectors are transformed into features through activation function ReLU, defined as Equation (8).
*ReLU*(*x*) = max(0, *x*)(8)

To match the output of this model, we then resize the final output vector back to the size of the input feature map for subsequent density map output, as defined by Equation (9).
(9)$$F_{SA}=VIEW({ReLU(SA}_{n})),n=1\ldots N$$

The function *VIEW*() is used to resize to the size of ***F****_d_*; ***F****_SA_* is the self-attention feature map, which is also the final output of this block.

## 4. Model Training and Testing

In this paper, we utilize two publicly available datasets to train and evaluate the accuracy of object counting in crowded scenes, with the main objects being vehicles and crowds. The TRANCOS dataset [35] is used for counting vehicles in extremely congested road images, while the ShanghaiTech dataset [36] is used for counting crowds in images from large-scale events with extreme congestion. We first introduce the evaluation metrics and datasets used in the experiments. The mean absolute error (MAE) and root mean square error (RMSE) are used as evaluation metrics for object counting accuracy, as defined by Equations (10) and (11), respectively.
(10)$$MAE=\frac{1}{N}\sum_{i=1}^{N}\left|Y_{i}-\hat{Y_{i}}\right|$$
(11)$$RMSE=\sqrt{\frac{1}{N}\sum_{i=1}^{N}{\left(Y_{i}-\hat{Y_{i}}\right)}^{2}}$$

Here, *N* represents the number of test images, *Y_i_* represents the true count of vehicles or people in the *i*-th frame, $\hat{Y_{i}}$ represents the predicted count by the model for vehicles or people in the *i*-th frame, and the total count of vehicles or people in the entire frame can be obtained by directly summing the entire density map.

The TRANCOS database consists of 1244 annotated images, with 870 images used for training and totaling 46,796 marked vehicles, all depicting scenes of extreme traffic congestion. We observed that some scenes were excessively blurred, so we applied histogram equalization to all images as preprocessing, then averaged them with the original images, as shown in Figure 7. It is evident that the images after histogram equalization are clearer than the originals, resolving issues such as blurriness and overexposure caused by low resolution, thus potentially enhancing the accuracy of crowded object counting.

We also found that the TRANCOS database did not fully annotate all vehicles in the images. Therefore, we re-annotated all images in the dataset to ensure that all vehicles in the images were included in the training, as illustrated in Figure 8. To maintain objectivity in comparison with other methods, we treated the results of this re-annotation as a new database, dividing it into the original TRANCOS and the re-annotated TRANCOS, and conducted separate experiments for analysis.

The second dataset is the ShanghaiTech Crowd Density Dataset, which comprises 1198 annotated images, totaling 330,165 people. The dataset is divided into two parts, A and B. Part A consists of 482 images, with 300 used for training, as depicted in Figure 9a. Part B consists of 716 images, with 400 used for training and the remaining used for testing, as shown in Figure 9b. The crowds in Part A are denser and contain more people compared to Part B, with varying image sizes.

## 5. Experimental Results

### 5.1. Crowded Vehicle Image Counting

The crowded vehicle image counting experiment calculates the vehicle count by summing the predicted density maps generated by the model, subtracting them from the ground truth values to obtain the error. An example of the experimental results is shown in Figure 10. Figure 10a illustrates a crowded vehicle image, and Figure 10b shows the detected density map overlaid with transparency onto the original images. It can be seen from Figure 10b that the brighter density map areas show areas where vehicles are densely packed. Here, we utilize the re-annotated TRANCOS database to analyze the accuracy of crowded object counting techniques. The channel depth of the feature map inputted to the self-attention module is set to 512, and the number of self-attention modules is set to 6.

Additionally, for comparison with other methods, we also include the average error from the original TRANCOS database in Table 1. We compare other models with our method, and it is evident that the use of self-attention indeed improves the accuracy of density estimation. Furthermore, we conduct the same experiments using the newly re-annotated TRANCOS database. The average error data are listed in Table 2. It is obvious that the error values decrease after re-annotation. This could be attributed to the accurate labeling of all vehicles in the images, reducing confusion caused by areas with vehicles not being labeled.

### 5.2. Crowd Density Counting

For the test images of crowded crowds in the ShanghaiTech database, the accuracy analysis calculation method involves subtracting the estimated density map from the ground truth to obtain the error value. The example of the experimental results is shown in Figure 11. Additionally, we compare our method with other crowd density counting methods, and the average errors are listed in Table 3. In part A, CSRNet [37] performs the best, while our proposed method ranks third in accuracy. However, in part B, our method achieves the best performance compared to other methods. Accuracy (ACC) is a metric commonly used to evaluate the performance of a classification model. It represents the ratio of the number of correct predictions to the total number of predictions made.

### 5.3. Ablation Study of Combining Self-Attention Mechanism

In this paper, we conduct a study on various parameters in combining self-attention modules to analyze their impact on accuracy and identify the optimal parameter combination. This analysis is crucial for enhancing the performance of the context-aware network when integrating self-attention modules. The parameters under investigation include the position and number of self-attention modules, as well as the patch size when vectorizing feature maps. For systematic analysis, both training and testing are based on the TRANCOS vehicle density dataset.

#### 5.3.1. Analysis of Object Counting Accuracy by Combining Self-Attention Modules at Different Positions

Regarding the insertion position of the self-attention module, we analyze the accuracy of crowded object counting for the three insertion positions, as shown in Figure 12. Position 1 is after VGG16, position 2 is after the multi-scale difference module, and position 3 is the location after connecting the first two positions. As shown in Figure 12, these positions correspond to applying self-attention feature enhancement to the ***F***_v_, ***F***_d_, and [***F***_v_|***F***_d_] features, respectively.

The experimental results, as shown in Table 4, clearly indicate that the accuracy metrics for Position 2 are the best. This is because the ***F***_d_ features, which have been enhanced by the self-attention mechanism, are derived from the fusion of multiple different scale features, resulting in better object density map features. Position 1, which only utilizes the basic VGG16 image feature extraction, has lower accuracy compared to Position 2. Position 3, on the other hand, shows a decrease in accuracy due to the confusion caused by the concatenation of the first two positions, which negatively affects the self-attention mechanism.

#### 5.3.2. Analysis of Object Counting Accuracy with Different Numbers of Self-Attention Modules

This paper also investigates the impact of the number of concatenated self-attention modules on object counting accuracy. We conducted experiments with 1, 3, 6, 9, and 12 self-attention modules, all placed at Position Two. The experimental results, as shown in Table 5, indicate that as the number of self-attention modules increases from 1 to 12, the error values gradually decrease, and the accuracy improves. However, although the error slightly decreases when using 12 modules compared to 6 modules, the significant increase in the number of parameters results in longer training times. Therefore, this study opts to use 6 concatenated self-attention modules for the model design.

#### 5.3.3. Analysis of the Impact of Patch Size on Object Counting Accuracy

The patch size determines the vector length input into the self-attention module. The longer the vector length, the more memory capacity is required. Assuming the feature map size is 30 in length, 40 in width, and 512 in depth, if the patch size is set to 10 × 10, the vector length will be (10 × 10 × 512), and there will be 3 × 4 vectors. If the patch size is reduced to 1 × 1, the required memory for training decreases, given the same feature map size. We tested three patch sizes: 15 × 20, 10 × 10, and 1 × 1. Due to memory constraints, the number of self-attention modules was kept at 1, and the object counting accuracy analysis was conducted at Position 2. The experimental results, as shown in Table 6, indicate that reducing the patch size also decreases the error value.

## 6. Conclusions

In this paper, the main contribution is to study how to combine the multi-head self-attention modules into the context-aware network. This study is crucial for enhancing the performance of the context-aware network when integrating self-attention modules. The parameters under investigation include the position and number of self-attention modules, as well as the patch size when vectorizing feature maps. Once the integration method is determined, we evaluate the counting accuracy on the TRANCOS database, the re-annotated TRANCOS database, ShanghaiTech Part A, and Part B, achieving accuracy rates of 85.9%, 90.0%, 83.4%, and 92.6%, respectively. Compared to the context-aware network, which serves as the primary architecture, we observe improvements of 0.5%, 1.1%, 0.4%, and 0.8% in crowded object counting accuracy. Based on the observation of the images in the TRANCOS database, we observed that some blur scenes can affect the counting accuracy. Hence, resolving issues such as blurriness and overexposure caused by low resolution still remains a challenge.

## Figures and Tables

**Figure 1 sensors-24-06612-f001:**
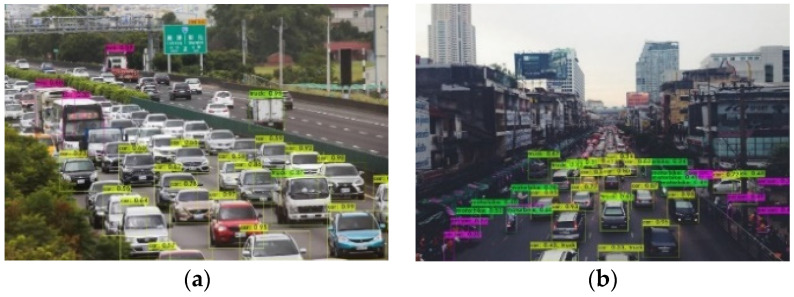
(**a**) Severe occlusion between vehicles causing object detection failure. (**b**) Vehicles in the far distance from the camera resulting in object detection failure.

**Figure 2 sensors-24-06612-f002:**
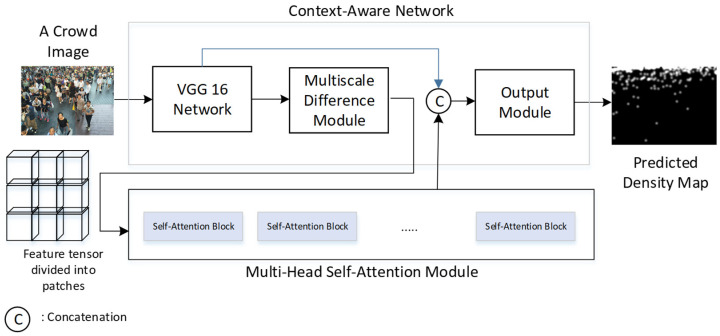
Model for counting crowded objects combining the context-aware network and self-attention mechanism.

**Figure 3 sensors-24-06612-f003:**
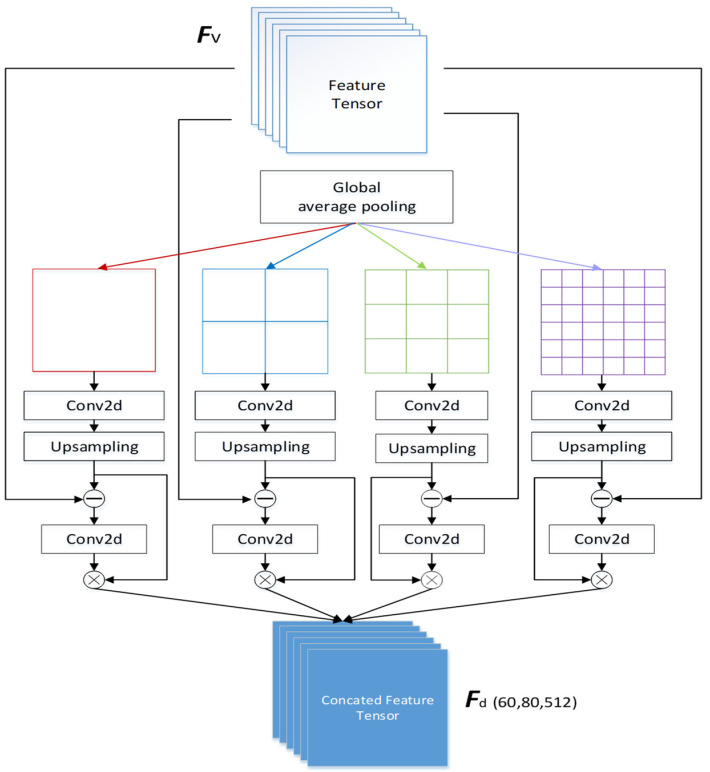
Network architecture of the multiscale difference block for feature fusion [25].

**Figure 4 sensors-24-06612-f004:**
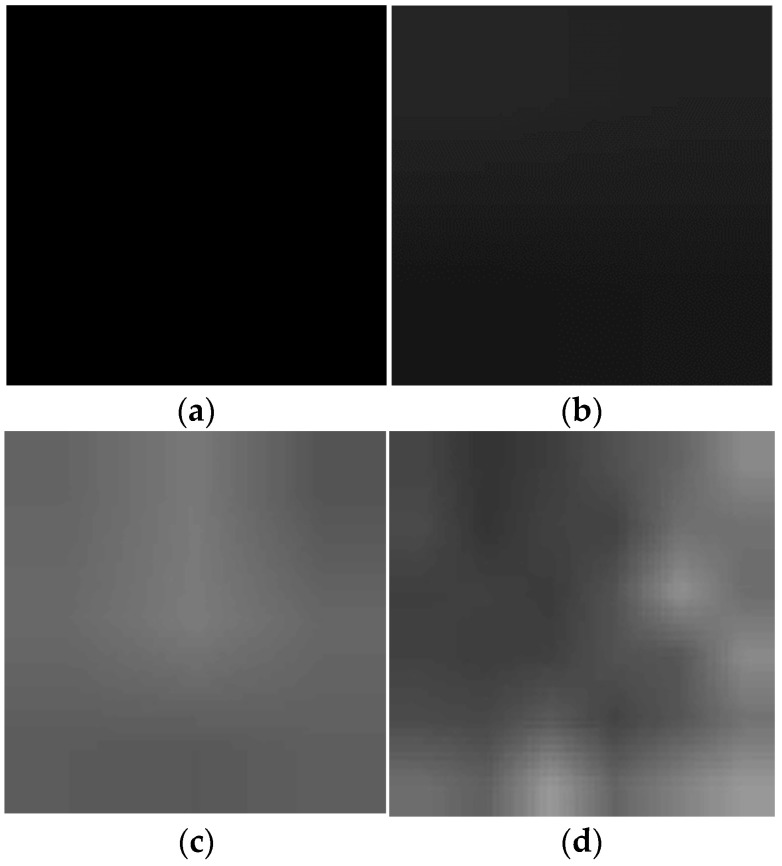
Feature maps after upsampling for each branch in the multiscale feature difference module. Feature map of the (**a**) 1 × 1 branch, (**b**) 2 × 2 branch, (**c**) 3 × 3 branch, (**d**) 6 × 6 branch.

**Figure 5 sensors-24-06612-f005:**
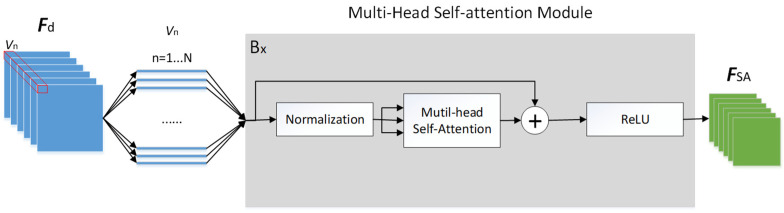
Integration of context-aware network and self-attention mechanism.

**Figure 6 sensors-24-06612-f006:**
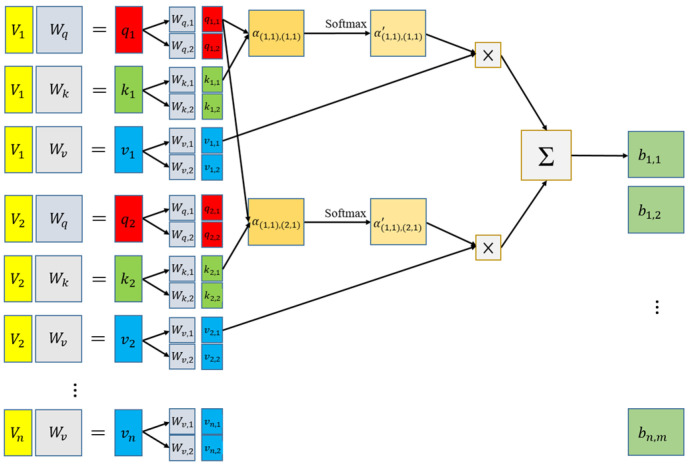
Schematic diagram illustrating the computation process of the multi-head self-attention method (using head = 2 as an example).

**Figure 7 sensors-24-06612-f007:**
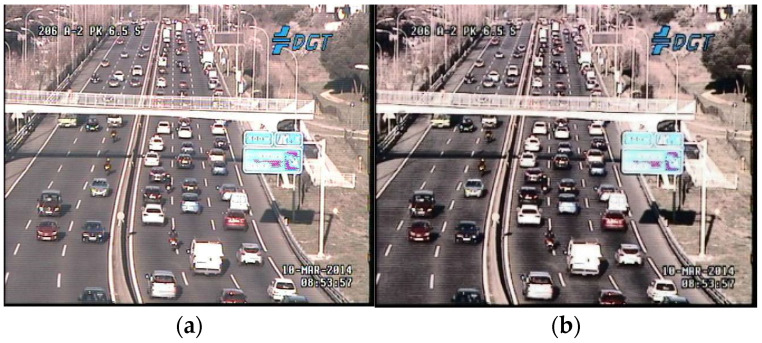
(**a**) Original image from the database. (**b**) Result after applying histogram equalization to the original image.

**Figure 8 sensors-24-06612-f008:**
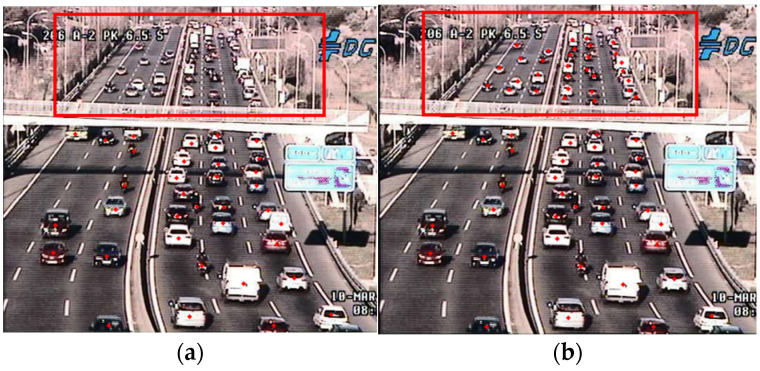
(**a**) Unmarked portion in the original image from the database. (**b**) Result after re-annotating the original image manually.

**Figure 9 sensors-24-06612-f009:**
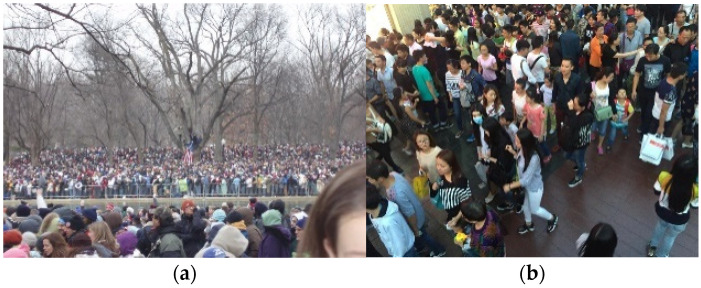
(**a**) Example image from the ShanghaiTech part A database. (**b**) Example image from the ShanghaiTech part B database.

**Figure 10 sensors-24-06612-f010:**
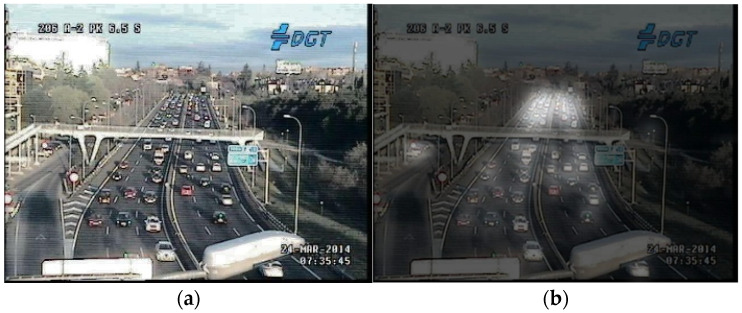
(**a**) A crowded vehicle image in TRANSCOS dataset. (**b**) Detected density map overlaid with transparency onto the original images for ease of observation.

**Figure 11 sensors-24-06612-f011:**
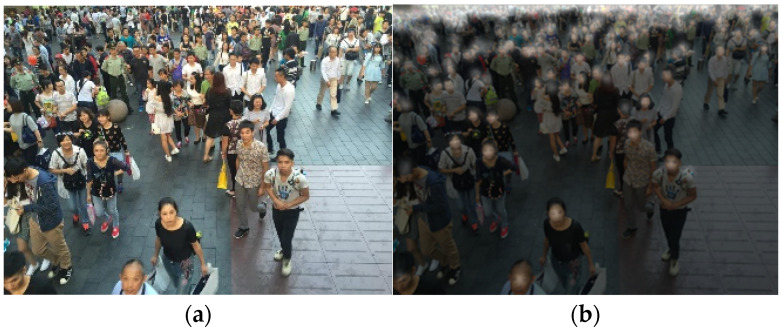
(**a**) Experimental example of crowded crowd input image from ShanghaiTech database. (**b**) The detected density map overlaid with transparency onto the original image.

**Figure 12 sensors-24-06612-f012:**
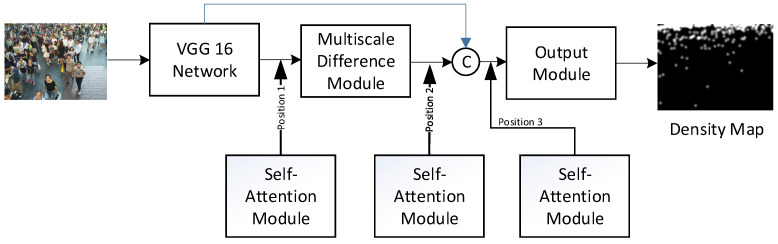
Analysis of object counting accuracy by combining self-attention module at different positions.

**Table 1 sensors-24-06612-t001:** Vehicle counting results from the original TRANCOS dataset.

Model	MAE	RMSE	Accuracy
Hydra [19]	10.99	-	72.24%
Context-Aware Network [25]	5.80	8.07	85.34%
OURS	5.57	7.11	85.93%

**Table 2 sensors-24-06612-t002:** Vehicle counting results from the re-annotated TRANCOS dataset.

Model	MAE	RMSE	Accuracy
Hydra [19]	9.73	-	80.46%
Context-Aware Network [25]	5.54	7.08	88.87%
OURS	4.96	6.62	90.04%

**Table 3 sensors-24-06612-t003:** Comparison of crowd counting methods on the ShanghaiTech dataset.

	Part_A			Part_B		
Method	MAE	RMSE	ACC	MAE	RMSE	ACC
ACSCP [38]	75.7	102.7	82.5%	17.2	27.4	86.0%
Liu et al. [39]	73.6	112.0	83.0%	13.7	21.4	88.9%
D-ConvNet [40]	73.5	112.3	83.0%	18.7	26.0	84.8%
IG-CNN [41]	72.5	118.2	83.2%	13.6	21.1	88.9%
IC-CNN [42]	68.5	116.2	84.1%	10.7	16.0	91.3%
CSRNet [37]	68.2	115.0	84.2%	10.6	16.0	91.4%
CAN [25]	73.7	117.5	83.0%	10.1	18.4	91.8%
OURS	71.6	116.8	83.5%	9.1	16.8	92.6%

**Table 4 sensors-24-06612-t004:** Analysis of accuracy with self-attention module inserted at different positions.

Position	MAE	RMSE	Accuracy
1	6.70	8.58	86.5%
2	4.94	6.51	90.0%
3	13.37	17.06	73.1%

**Table 5 sensors-24-06612-t005:** Analysis of object counting accuracy with different numbers of self-attention modules.

Number of Self-Attention Module	MAE	RMSE	Accuracy
1	5.27	6.91	89.4%
3	5.21	6.88	89.5%
6	4.97	6.63	90.0%
9	4.96	6.58	90.0%
12	4.94	6.51	90.1%

**Table 6 sensors-24-06612-t006:** Analysis of the impact of patch size on object counting accuracy.

Patch Size	MAE	RMSE	Accuracy
15 × 20	6.36	8.05	87.2%
10 × 10	5.31	6.86	89.3%
1 × 1	4.97	6.63	90.0%

## Data Availability

Data are contained within the article.

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
