# Peer review of "A Crowded Object Counting System with Self-Attention Mechanism"

_sensors, 2024, doi:10.3390/s24206612_

Round 1

Reviewer 1 Report

Comments and Suggestions for Authors

The manuscript entitled A Crowded Object Counting System with Self-Attention Mechanism has the potential implication for detecting dense vehicles accurately. However, to meet the journal standard, some notes should be considered by the authors:

1. Part 1 and 2 are lack of the references. I think to improve the novelty, the authors need add more references and it should be up to date. 

2. How the authors explain the combination of the Context-Aware network architecture with the self-attention mechanism of the Vision Transformer has accuracy 92.6%? the discussion should be improved carefully by analyze and compared with the other results.

3. Could the authors explain the difficulties to get higher accuracy? is the main factor the same with others research?

Comments on the Quality of English Language

English should be improved. I suggest the authors carefully build the sentences and use the punctuation.

Author Response

1. Part 1 and 2 are lack of the references. I think to improve the novelty, the authors need add more references and it should be up to date. 

2. How the authors explain the combination of the Context-Aware network architecture with the self-attention mechanism of the Vision Transformer has accuracy 92.6%? the discussion should be improved carefully by analyze and compared with the other results.

3. Could the authors explain the difficulties to get higher accuracy? is the main factor the same with others research?

Reviewer 2 Report

Comments and Suggestions for Authors

In this work, the Context-Aware Network architecture and the self-attention mechanism are employed to propose a new model for counting crowded and dense objects. The work is well-written, well-organized, and additional work was carried out with the reannotation of the TRANCOS Database. However, some questions need to be answered, and minor details need to be corrected.

The proposed tools, Context-Aware Network architecture and the self-attention mechanism, already exist, so the innovation lies in the integration and application of these algorithms together in a novel context.

The equations, symbols, and variables need to be revised exhaustively because they differ in the equations from those in the text. For example, in equation (1) the function GP_ave appears described in line 148 in bold symbols, while in equation (1) it does not appear similarly. The same situation is repeated in line 153 with S_j, and F_v, in equation (2) with S_j, in line 169 with W_j, etc.

The superscripts in lines 188 and 189 are misplaced for the variables D_i^gt and D_i^est.

​In line 205, the authors do not define the variable MHA in equation (7); I suppose it is Multi-Head Attention. In the same equation (7), the variable LN is not defined.

Although the abbreviation ReLU is well-known, I suggest defining it.

The experimental results should be extended for the TRANCOS database with a comparison to several methods and different metrics. Showing the ROC curves could be interesting. The ShanghaiTech dataset is more complete; however, the results do not provide a significant advantage over the proposed method.

In line 333, the authors wrote: "This is likely because the Fd features, which have been enhanced by the self-attention mechanism, are derived from the fusion of multiple different scale features, resulting in better object density map features." First, the word "likely" gives the impression that the authors do not know what is happening with their evidence. Second, the variable F_d requires correction.

The number of references is insufficient and needs to be updated to more recent years.

Comments on the Quality of English Language

The english quality requires minor editing. 

Author Response

The equations, symbols, and variables need to be revised exhaustively because they differ in the equations from those in the text. For example, in equation (1) the function GP_ave appears described in line 148 in bold symbols, while in equation (1) it does not appear similarly. The same situation is repeated in line 153 with S_j, and F_v, in equation (2) with S_j, in line 169 with W_j, etc.

The superscripts in lines 188 and 189 are misplaced for the variables D_i^gt and D_i^est.

​In line 205, the authors do not define the variable MHA in equation (7); I suppose it is Multi-Head Attention. In the same equation (7), the variable LN is not defined.

Although the abbreviation ReLU is well-known, I suggest defining it.

The experimental results should be extended for the TRANCOS database with a comparison to several methods and different metrics. Showing the ROC curves could be interesting. The ShanghaiTech dataset is more complete; however, the results do not provide a significant advantage over the proposed method.

In line 333, the authors wrote: "This is likely because the Fd features, which have been enhanced by the self-attention mechanism, are derived from the fusion of multiple different scale features, resulting in better object density map features." First, the word "likely" gives the impression that the authors do not know what is happening with their evidence. Second, the variable F_d requires correction.

The number of references is insufficient and needs to be updated to more recent years.

Reviewer 3 Report

Comments and Suggestions for Authors

The paper develops crowded object counting system using density estimation. The purpose of this research is object detection, when the objects are small, crowded, and dense. Here are some points which need to be incorporated before publication of the paper.

·         The abstract is properly structured but here DNN comes directly without its definition as it makes confusion for the readers in its first appearance.

·         As the Title is “A Crowded Object Counting System with Self-Attention Mechanism”, while the introduction section is directly started from define YOLO series. The start of the introduction mismatch with the title, and it also lack information like overviewing the Title, discussing the impact of the specific field for research area, advantages, disadvantages, research contributions, and motivation.

·         The contributions must be added at the end of the introduction section in bullet form.

·         In Figure 2, "Conv" and "Upsample" are underlined in red which seem unprofessional. Additionally, the model illustrated in Figure 2 appears to be derived from the architecture developed by reference 5. Upon reviewing the proposed model architecture in Figure 4, it is presented in a similar manner. To avoid confusion and ensure clarity, it would be beneficial to revise Figure 2 to differentiate it more distinctly from the previous model depiction, thereby avoiding the perception of direct replication.

·         While training with the TRANSOS dataset the ratio of training and testing set is not mentioned.

·         A robust model should perform well across a range of object sizes and densities, not just when using a specific patch size. This means it should be capable of accurately detecting objects whether they are large, small, or densely clustered together. Table 6 mentioned the increasing accuracy along with the decrease of patch size. So, this limitation must be added in the end while concluding the paper.

·         As it mostly based on Object detection, a recent work on object detection can be included in the paper https://doi.org/10.1109/TII.2021.3116377

·         The conclusion effectively summarizes the key findings and contributions of your study; however, it would greatly benefit from further elaboration to provide a more comprehensive closure. Expanding on the practical implications of your improved accuracy rates in crowded object counting would enhance the significance of your research. Additionally, discussing how these findings could impact real-world applications or suggesting potential future research directions would enrich the conclusion, providing a clearer roadmap for readers and researchers in the field. This expansion would not only strengthen the conclusion but also highlight the broader impact and potential of your work.

Comments on the Quality of English Language

The English need improvement in terms of professionalism and connection.

Author Response

The abstract is properly structured but here DNN comes directly without its definition as it makes confusion for the readers in its first appearance.

  • As the Title is “A Crowded Object Counting System with Self-Attention Mechanism”, while the introduction section is directly started from define YOLO series. The start of the introduction mismatch with the title, and it also lack information like overviewing the Title, discussing the impact of the specific field for research area, advantages, disadvantages, research contributions, and motivation.
  • The contributions must be added at the end of the introduction section in bullet form.
  • In Figure 2, "Conv" and "Upsample" are underlined in red which seem unprofessional. Additionally, the model illustrated in Figure 2 appears to be derived from the architecture developed by reference 5. Upon reviewing the proposed model architecture in Figure 4, it is presented in a similar manner. To avoid confusion and ensure clarity, it would be beneficial to revise Figure 2 to differentiate it more distinctly from the previous model depiction, thereby avoiding the perception of direct replication.
  • While training with the TRANSOS dataset the ratio of training and testing set is not mentioned.
  • A robust model should perform well across a range of object sizes and densities, not just when using a specific patch size. This means it should be capable of accurately detecting objects whether they are large, small, or densely clustered together. Table 6 mentioned the increasing accuracy along with the decrease of patch size. So, this limitation must be added in the end while concluding the paper.
  • As it mostly based on Object detection, a recent work on object detection can be included in the paper https://doi.org/10.1109/TII.2021.3116377
  • The conclusion effectively summarizes the key findings and contributions of your study; however, it would greatly benefit from further elaboration to provide a more comprehensive closure. Expanding on the practical implications of your improved accuracy rates in crowded object counting would enhance the significance of your research. Additionally, discussing how these findings could impact real-world applications or suggesting potential future research directions would enrich the conclusion, providing a clearer roadmap for readers and researchers in the field. This expansion would not only strengthen the conclusion but also highlight the broader impact and potential of your work.
